# Similarity-Driven Regularization for Aligning Chemical and Latent Spaces in Molecular Design

## Abstract

Generative models play a pivotal role in molecular design by effectively generating target molecules. Among these, generative models with latent space stand out due to their robust latent space representation capabilities, powerful dimensionality reduction ability and controllability of generation. In molecular design applications, generative models with latent space convert input molecules into latent variables, capturing essential molecular features including both structural and property-related characteristics. Ideally, similar molecules should map to proximate latent variables. However, previous studies have shown an inconsistency between molecular similarity in the chemical space and that in the latent space. This inconsistency will impede the accurate representation and complicate subsequent design process,such as leading to higher optimization budget. To address this, we propose Molecular Similarity-Aware Consistency Regularization (MSCR), a straightforward regularization approach aimed at preserving the consistency of molecule similarity. Our method proposes a brief but effective regularization technique to align chemical space and latent space,clearly reflect similarity relationships in latent space. We leverage Matched Molecules Pairs (MMPs) to introduce more robust similarity information than other conventional augmentation methods. Extensive experiments demonstrate that MSCR not only maintains molecules pairs similarity but also enhance optimization performance in molecular latent space tasks, without additional costs. Furthermore, our visualizations highlight molecular inconsistencies, thus underscoring the significance of our approach and improving the interpretability and relevance of our work.

## 1 Introduction

Molecular design is a complex process about how to design molecules with specific properties and functions in order to solve practical problems. Vast and complicated chemical space makes it challenging to directly identify suitable molecules. Generative model with latent space is a novel and principled paradigm. Molecules are mapped to continuous latent space and represented by latent variables. While a molecule is transformed to a variable, the whole design space is transformed from complicated discrete space to relatively simple continuous space. From this point of view, this transformation is simplification for hard molecular design problems. Compared to discrete problems, continuous problems usually can be solved easier because they usually have smooth solutions spaces and the objective functions and constrains in continuous problems are often easier to model and analyze. In this case, lots of remarkable approaches can be used (i.e Bayesian Optimization and gradient descent) directly. What's more, molecular design process will have more interpretability and controllability if the whole process base on latent space (Vahdat et al. (2021); Yi et al. (2020); Nie et al. (2021)). In such space, improvement of molecular properties can be observed clearly (Prykhodko et al. (2019); Winter et al. (2019)), molecular edit can be reflected regularly (Blaschke et al. (2018)) and so on.

Generative model can be used to generate diverse and high-quality instances. Briefly, the distribution of desired data in latent space will match a specific and simple distribution (e.g. Gaussian distribution) through training. In this case, it is easy to get adequate desired data by sampling from the distribution. Typically, A generative model with latent space consists of an embedding process

and a restoring process. For example, variational autoencoders(VAEs) utilize an encoder to embedding origin data to latent variables and recover the data with a decoder vice versa. Generator and discriminator in generative adversarial networks (GANs) and adding noise and denoising process in diffusion models (DMs) play the similar role. Once trained, the embedding module of generative model with latent space can be used to obtain low-dimension representation of data by latent variables, and the data can be reconstructed based on its corresponding latent variable by the restoring module. There are various generative model with latent space used in latent space molecular design areas. These work has achieved promising results for generating desirable molecules and editing molecules flexibly. Usually, existing work base on a paradigm, initial molecules are given to embedding module and their latent variable can be gain. And then using gradient descent, Bayesian optimization or other wonderful methods to get desired latent variables of molecules. Finally, the desired molecules can be obtained with restoring module based on chosen latent variables. For such an paradigm, an important premise is the need for high-quality latent spaces. A high-quality latent space means that latent variables in it can represent molecules exactly, the space should include molecules et cetera. If the latent variable can't represent the molecule exactly, the follow-up process is meaningless because these design operations work on another molecule. If latent space is small and neglects a lot potential molecules, it is hard to discover appropriate molecules and make detailed edit. Thus,the quality of latent variable is essential for the performance of these model.

Many researches have paid attention to improve quality of latent space in generative model with latent space specially in VAEs, often handling the so-called latent variable collapse problem-where the approximate posterior distribution induced by the encoder collapses to the prior over the latent variables (He et al. (2019); Lucas et al. (2019)). However, these settings can not improve the quality of molecular latent space effectively, because these approaches fail to consider the relative relations between different molecules. When mapping chemical space to latent space, both individual molecular representation and the relationship between different molecules should be taken into consideration. Based on Quantitative Structure-Activity Relationship (QSAR) (Tropsha (2010); Cherkasov et al. (2014)), properties of a molecule depend on its structure strongly. Thus, starting from an original molecule to find structurally similar molecules with improved properties is indeed an ideal paradigm in molecular design, as it allows for leveraging existing knowledge and data to enhance the success rate and efficiency of research and development. Consequently, it is necessary to maintain similarity information. In other words, assuming a high-quality molecular generative model with latent space, a pair of similar molecules should have similar latent variable in latent space. It means molecular latent space should have consistency with chemical space in similarity. We define this consistency as molecular similarity consistency (MSC). MSC is important for molecular design, especially for molecular optimization. A generative model with latent space that integrates MSC can easily find molecules with good properties and high similarity.

Figure1 (a) illustrates the phenomenon existing in generative models with latent space. We present three molecules in chemical space and latent space, where two of them are similar and the other is dissimilar. Notably, the two similar molecules do not appear very similar in latent space, with one being more similar to the dissimilar molecule in chemical space. Such a phenomenon suggests that the current generative models with latent space do not capture the intrinsic similarity relationship between molecules very well. Figure1 (b) shows that such inconsistencies are common, and we have counted the chemical space similarity and latent space similarity of several methods and found that there is a huge gap between these two similarities, which suggests that many of the similarity relationships are broken during the mapping process of the generative model with latent space.Based on the above simple experiments, we can demonstrate that the current generative model with latent space has many shortcomings in MSC, which presents challenges for downstream molecular tasks.

In this paper, we propose a powerful similarity-relevant regularization technique to retain the MSC. This approach addresses the goals from both distribution and metric perspectives. From the distribution perspectives, we ensure that similar molecules have comparable distributions in the latent space. From the metric perspectives, we align the similarity relationships both in chemical space and latent space. This regularization technique is plug-and-play and can be applied to any molecular generative model with latent space to maintain MSC and improve the quality of the latent space and the performance of molecular design performance. Because of the comprehensive understanding of similarity between molecules, we call our work Molecular Similarity-Aware Consistency Regularization (MSCR). There are three main contributions in our work:

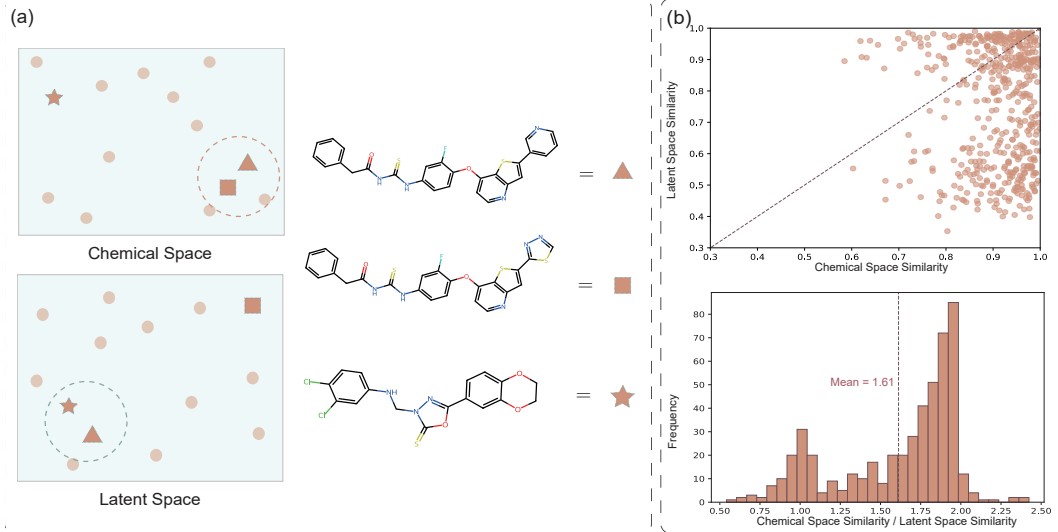

Figure 1: The phenomenon of molecular similarity inconsistency. (a) Two molecules that are similar in chemical space are represented by triangles and squares, respectively, as well as a dissimilar molecule represented by a star. But when mapped into latent space, the dissimilar molecule instead becomes more similar to one of the similar molecules as a pair of molecules. (b) The above figure represents the correspondence between chemical space similarity and latent space similarity. The x-axis indicates the chemical space similarity of a pair of molecules, while the y-axis represents the latent space similarity of the same pair. We plotted a line y = x to represent the ideal correspondence. The figure also shows that the current latent space does not maintain a good consistency in similarity.The figure below illustrates the statistical relationship between chemical space similarity and latent space similarity. The x-axis represents the ratio of these two similarities. Ideally, the ratio should be as close to 1 as possible; however, the data shown in the figure indicates a certain deviation from 1. This suggests that the similarities in these two spaces are inconsistent.

- We have designed a new loss component, the similarity loss, which can be easily integrated into existing models. This plug-and-play loss function ensures that molecular similarity is preserved. By maintaining consistency between chemical space and latent space, it effectively aligns molecular similarity in chemical space and latent space.

- We use a new data augment approach MMPs to get similar molecular pairs. MMPs data effectively captures fine-grained differences between molecules. This nuanced understanding allows for more precise optimization in drug design and aids in identifying critical features that contribute to desired effects.

- In our experiments, we apply the proposed approach to different generative models with latent space.We find MSCR can improve the quality of latent space and get better performance than their based model. What's more, our model can achieve better optimization performance with simple Bayesian Optimization and gradient descent. We also find clear optimization path to prove our model's interpretability.

## 2 RELATED WORK

**generative model with latent space in Molecular Design.** In order to improve generative modeling ability, extensive research has been devoted to the development of more expressive molecular generative model with latent space. Variational Autoencoders(VAEs) is a powerful approach way including a encoder, a decoder and a pre-defined prior distribution to design molecules. VQ-VAEs (Chen et al. (2021); Xia et al. (2023)) proposes to quantize latent variables and use a VAE autogressive model to learn a prior. Notin et al. (2021) uses VAEs to learn the distribution of molecules and sample desirable molecules to optimize molecules in several tasks. JT-VAE (Jin et al. (2018))

uses junction tree to describe a molecule and use junction tree and graph simultaneously to generative high-quality molecules. Generative Adversarial Networks(GANs) can generate high-quality molecules by cyclic interactions of generators and discriminators. MolGAN (De Cao & Kipf (2018)) utilizes GANs directly on graph-structured molecules data and combine the model with a reinforcement learning to guide the generation of molecules with specific desired chemical properties. LatentGAN (Afzali et al. (2021)) uses an autoencoder and a GAN to generative high-quality and it can perform well both in generating random drug-like compounds and target-biased compounds. GFlowNet (Bengio et al. (2021)), based on a view of the generative process as a flow network, regards molecular generative process as a set of trajectories adding atoms. HN-GFN (Zhu et al. (2024)) leverages the hypernetwork-based GFlowNetsas an acquisition function optimizer and sample a diverse batch of candidate molecular graphs from an approximate Pareto front. Except the previous work focusing on generating molecules as 2D graphs and sequences. Recent years witness a lot of methods generating molecules in 3D space. GEOLDM (Xu et al. (2023)) composes of autoencoders encoding structure into continuous latent codes and diffusion model operating. It achieves a great performance in generating biomolecules and controllable generation. Despite the existing generative model with latent space can generate desirable molecules, these methods can hardly take the similarity information into consideration.This makes the distribution of latent space haphazard, with a mixed bag of molecular masses in each region, which creates a lot of extra burden for search and generation. It is possible to identify molecules of a higher quality, but this is accompanied by the discovery of molecules of a lower quality.

**Consistency Regularization.**Consistency regularization was widely used in semi-supervised learning. Its goal is to make representations less sensitive to image transformations, improving the classification of unlabeled images. Consistency is typically achieved by minimizing the L2 distance between the classifier's output and the output of its semantically preserved transformation, or by minimizing the KL divergence between the classifier's label distribution and its transformed label distribution. In recent years, Consistency regularization is used on generative model to improve it performance. Consistency regularization has been applied to Generative Adversarial Networks (GANs) (Zhang et al. (2019)). Indeed Wei et al. (2018) and Zhao et al. (2021) show that applying consistency regularization on the discriminator of GANs. What's more, consistency regularization is applied to Variational AutoEncoders (VAEs). Denoising auto-encoder(DAEs) (Vincent et al. (2008; 2010)) corrupt an image $x$ to $x'$ by adding Gaussian noise and minimizes the distance between the reconstruction $x'$ and the image $x$. The inspiration of the work is to learn representations that are not sensitive to noises. Besides, Contractive auto-encoder(CAEs) Rifai et al. (2011) uses norm constraint on the JacoBian forces the representations to make encoder be insensitive to tiny changes of its inputs. CR-VAEs (Sinha & Dieng (2021)) minimizes KL divergence to reduce the sensitivity of encoder to image corrupt. Similar consistency regularization is also used in diffusion model. Daras et al. (2024) proposes Consistency property(CP) to state that predictions of the model on its own generated data are consistent across time. MSCR are quite different with these methods. Our method focuses on molecules domain rather than images. The motivation of these methods is to improve the insensitivity of models, but our model is to improve the quality of the latent space and improve the efficiency of downstream task. Our method can add to any generative model with latent space rather than a specific model.

## 3 METHOD

In this section, we formally describe Molecules Similarity-Aware Consistency Regularization(MSCR). Our approach utilize a novel data augmentation method based on Matched Molecule Pairs(MMPs). We will introduce the detail of MMPs and the detailed augmented data processing methods in section 3.1. In section 3.2, we will introduce the MSCR from its distribution items and metric items. We show the framework in Figure2 (b). Finally, we briefly summarize the training scheme in section 3.3.

### 3.1 MATCHED MOLECULES PAIRS

Matched Molecules Pairs (MMPs) are a concept in medicinal chemistry and drug design. An MMP is a pair of molecules that differ by a single, well defined structural change, often involving the replacement of one chemical group with another. As a result, the core scaffold or main structure

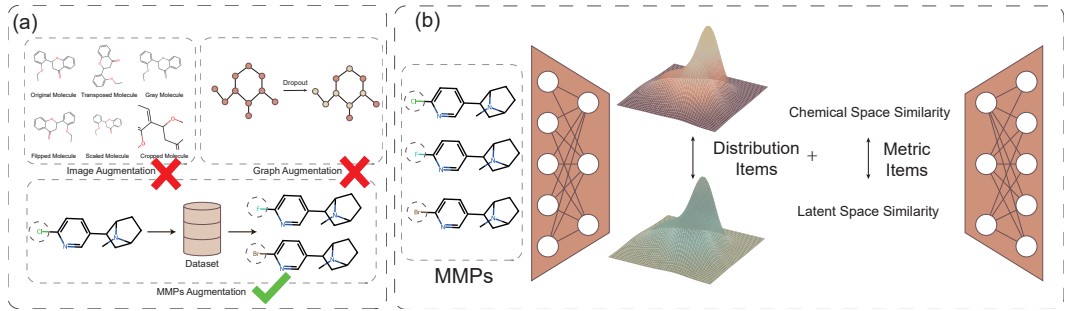

Figure 2: The approaches of MMPs and the framework of MSCR.(a) Other augmentation methods often lead to ineffective and redundant molecular information as shown in the figure, and they typically do not provide new insights. In contrast, MMPs are based on chemical rules and provide the model with a wealth of useful information.(b) MSCR introduces distribution items and metric items together to capture the similarity relationships between molecules, aligning the similarities in chemical space and latent space.

of the molecules remains the same or very similar, with only a minor localized difference. Due to the small structural change, these molecules often share many similar physicochemical properties, such as molecular weight, hydrophobicity, and polarity. Additionally, if the structural change has a minimal impact on biological activity or pharmacokinetics, the two molecules might also exhibit similar biological properties. Generally, MMPs tend to produce molecules that are similar in both structure and properties.

In our work, we use MMPs as an approach of data augmentation. Firstly, MMPs introduce an effective and plug-in-play way to get similar molecules and are suitable to general molecular representation method(e.g. Sequence, graph and image). Secondly, MMPs-based data augmentation introduces subtle but meaningful variation in the dataset making it include fine-grained similarity data(molecules in a MMP) and coarse-grained similarity data(molecules in different MMPs). Such setting benefits to improve diversity of dataset and generalization of model. Thirdly, MMPs are based on chemical knowledge, ensuring that the augmented data is relevant and realistic. In other words, the new molecules and their properties are scientifically grounded, meaning they are not just random or arbitrary changes. This is crucial for ensuring that the data augmentation process yields valuable and applicable results. In contrast, other augmentation methods do not provide this guarantee. For example, sequence data is typically augmented through random deletion, random insertion, etc.; image data through flipping, rotation, scaling, and cropping; and graph data through node dropping, edge dropping, and so on. If these methods were applied directly to molecular data augmentation, they could easily result in invalid or meaningless molecular structures. To address these issues, MMPs approach can easily provide the valid and meaningful data. Figure2 (a) illustrates the detailed process of data augmentation utilizing MMPs. Given a dataset, we identify the corresponding MMP data for each entry based on established MMP rules.

## 3.2 MOLECULAR SIMILARITY-AWARE CONSISTENCY REGULARIZATION

We consider a latent space model $p_\theta = p_\theta(x|z) \cdot p(z)$, where $x$ denotes an observation and $z$ is its corresponding latent variables. The marginal $p(z)$ is a prior over the latent variable and $p_\theta(x|z)$ is an exponential family distribution whose natural parameter is a function of $z$ parameterized by $\theta$ (e.g. a neural network). Our goal is to learn the parameters $\theta$ and a posterior distribution over the latent variables.

### 3.2.1 DISTRIBUTION ITEMS

Consider a MMP-based augmentation of $a(x'|x)$ on molecule $x$. As mentioned before, the similar latent variable should be provided to $x$ and its MMP $x'$. And these variables are based on a specific distribution. Intuitively, by bringing the two distributions closer together, we position similar molecules closer in latent space. This approach is designed to enhance the robustness of the model

and promote a more efficient learning process. This method serves as a coarse-grained constraint that ensures spatial consistency from a broader viewpoint. Unlike more stringent restrictions, it allows us to retain potential latent variables longer, avoiding premature sacrifices in the modeling process. We use a regularization method that ensures consistency of the distribution consistency. We draw $x'$ from $a(x'|x)$ as follows:

$$x' \sim a(x'|x) \Longleftrightarrow \varepsilon \sim p(\varepsilon) \; and \; x' = g(x, \varepsilon). \tag{1}$$

Here $g(x, \varepsilon)$ is a substructure change of a molecule $x$. MSCR then minimizes

$$\mathcal{L}_{MSCR}(x, x') = \mathbb{E}_x \mathcal{L}_{origin}(x) + \mathbb{E}_{a(x'|x)} \mathcal{L}_{origin}(x') + \lambda \cdot \mathcal{C}_{Distribution}(x, x', \phi), \tag{2}$$

where $\mathcal{L}_{origin}$ denotes the original loss of corresponding generative model with latent space.The distribution consistency term $\mathcal{C}_{Distribution}(x, \phi)$ is

$$\mathcal{C}_{Distribution}(x, x', \phi) = \mathbb{E}_{a(x'|x)}[KL(q_\phi(z|x')||q_\phi(z|x))]. \tag{3}$$

Minimizing the objective in Eq.2 minimizes the likelihood of the molecules and their augmentations while enforcing distribution consistency through $\mathcal{C}_{Distribution}(x, \phi)$. Minimizing $\mathcal{C}_{Distribution}(x, \phi)$,which only affects the embedding module (with parameter $\phi$), forces the molecules and their augmentation to lie close to each other in the latent space. The hyper-parameter $\lambda$ controls the strength of this constraint and its positive or negative value depends on whether the data is a positive sample or a negative sample.

### 3.2.2 METRIC ITEMS

Similarity consistency can be maintained by distribution-relevant KL divergence term. However, it's hard to keep the detailed consistency because we think it is broader to think in terms of distributions, this only gives a rough constraint on the model. To further improve the portrayal of similarity consistency, we introduce the metric-relevant consistency items

$$\mathcal{C}_{Metric}(x, x', \phi) = \mathbb{E}_{a(x'|x)}|\frac{Sim_L(q_\phi(z|x'), q_\phi(z|x)))}{Sim_C(x', x)} - 1|, \tag{4}$$

where $Sim_L$ and $Sim_C$ represent the similarity measurement in latent space and chemical space. In our work, $Sim_L$ is cosine similarity and $Sim_C$ is Tanimoto Similarity. The standard cosine similarity range should be $[-1, 1]$. In particular, we normalize the cosine similarity to $[0, 1]$ in order to make the two similarity measurement in the same range.

Combining the distribution and metric consistency, MSCR minimizes

$$\mathcal{L}_{MSCR}(x, x') = \mathbb{E}_x \mathcal{L}_{origin}(x) + \mathbb{E}_{a(x'|x)} \mathcal{L}_{origin}(x') + \lambda_1 \cdot \mathcal{C}_{Distribution}(x, x', \phi) + \lambda_2 \cdot \mathcal{C}_{Metric}(x, x', \phi), \tag{5}$$

where $\lambda_2 \geq 0$ is another hyper-parameter and control the strength of metric consistency constraint.

### 3.3 TRAINING

With the proposed formulation and practical parameterization, we now present the training schemes for MSCR. While objectives for training the model are already defined in Eq.5, it is still unclear whether the two component should be trained one by one, or optimized simultaneously. Previous work about generative model with latent space (Sinha et al. (2021)) shows that the two-stage training strategy usually leads to the better performance, and we notice the similar phenomena in our experiments. This means we first train MSCR with vanilla loss, and then train the generative model with latent space on the latent restoring modules. A formal description of the training process is provides in Alg.3.3.

Some objectives may be intractable, but we can effectively approximate them using Monte Carlo methods. Monte Carlo approximation is powerful because it leverages the Law of Large Numbers, which states that as the number of random samples increases, the average of those samples converges to the expected value of the distribution. Specifically, we approximate the regularization term by sampling from the dataset.

---

**Algorithm 1** Molecular Similarity-Aware Consistency Regularization

---

1: **Input:** Molecule Dataset $X = (x, x', sim_C(x, x'))$, distribution consistency strength $\lambda_1$, and metric consistency strength $\lambda_2$
2: **Initial:** embedding module $q_\phi$ and restored module $p_\theta$
3: **First Stage:Original Loss Training**
4: **while** $\phi$ and $\theta$ have not converged **do**
5:     Draw minibatch of molecules$\{(x, x', sim_C(x, x')\}_{n=1}^B$
6:     Compute $\mathcal{L}_{origin}(x)$: $\mathbb{E}_x \mathcal{L}_{origin}(x) \approx \frac{1}{B}\sum_{n=1}^B (\mathcal{L}_{origin}(x))$
7:     Compute $\mathcal{L}_{origin}(x')$: $\mathbb{E}_{a(x'|x)} \mathcal{L}_{origin}(x') \approx \frac{1}{B}\sum_{n=1}^B (\mathcal{L}_{origin}(x'))$
8:     $\mathcal{L} = \mathcal{L}_{origin}(x) + \mathcal{L}_{origin}(x')$
9:     $\phi, \theta \longleftarrow optimizer(\mathcal{L}, \phi, \theta)$
10: **end while**
11: **Second Stage:Distribution and Metric Consistency Training**
12: Fix restoring module parameter $\theta$
13: **while** $\phi$ and $\theta$ have not converged **do**
14:     Draw minibatch of molecules$\{(x, x', sim_C(x, x')\}_{n=1}^B$
15:     Compute $\mathcal{L}_{origin}(x)$: $\mathbb{E}_x \mathcal{L}_{origin}(x) \approx \frac{1}{B}\sum_{n=1}^B (\mathcal{L}_{origin}(x))$
16:     Compute $\mathcal{L}_{origin}(x')$: $\mathbb{E}_{a(x'|x)} \mathcal{L}_{origin}(x') \approx \frac{1}{B}\sum_{n=1}^B (\mathcal{L}_{origin}(x'))$
17:     Compute $\mathcal{L}_{distribution}(x, x'), \mathcal{L}_{metric}(x, x')$
18:     $\mathcal{L} = \mathcal{L}_{origin}(x) + \mathcal{L}_{origin}(x') + \mathcal{L}_{distribution}(x, x') + \mathcal{L}_{metric}(x, x')$
19:     $\phi \longleftarrow optimizer(\mathcal{L}, \phi)$
20: **end while**
21: **return** $q_\phi, p_\theta$

---

# 4 EXPERIMENTS

In this section, we justify the advantages of MSCR with comprehensive experiments. Firstly, we introduce our experiment setup in section 4.1. Secondly, we report some experiment results and analysis on similarity consistency in section 4.2.Thirdly, we show some experiment results and analysis on Bayesian optimization,gradient descent and random sampling in section 4.3. Finally, we also provide further ablation studies in section 4.4.

## 4.1 EXPERIMENTS SETTING

**Baselines.** To evaluate our approach, we chose two generative models with latent space to demonstrate the tangible improvement of our approach on the quality of the latent space of the generative model.1) TransVAE (Dollar et al. (2021)) is a molecular generative model with latent space for molecules based on the transformer architecture. By combining the ATTENTION mechanism with the VAE prediction of mean and variance generation, the model learns a complex molecular syntax and achieves significant improvement in many downstream tasks.2) GEOLDM (Xu et al. (2023)) organically combines the autoencoder architecture with the diffusion model. GEOLDM builds a point-structured latent space to capture critical roto-translational equivariance constraints and has achieved great success on 3D molecular generation tasks.

**Datasets.** To demonstrate the generality of our approach, we tested it on both 2D molecule generation tasks and 3D molecules. Thus, for the 2D molecule generation task, we chose two widely used datasets, ZINC (Irwin & Shoichet (2005)) and ChEMBL (Gaulton et al. (2012)), for model training and testing. Following Jin et al. (2018), we use the ZINC molecule dataset from Kusner et al. (2017) for our experiments, with the same training/testing split. It contains about 250K drug molecules extracted from the ZINC dataset (Sterling & Irwin (2015)). For the ChEMBL dataset, we are consistent with Jin et al. (2020), which contains a total of 1.02 million training samples. For 3D molecular generation tasks, QM9 is one of the most widely used datasets in molecular machine learning research. QM9 contains 3D structures together with several quantum properties for 130k small molecules, limited to 9 heavy atoms (29 atoms including hydrogens). Following Anderson et al. (2019), we split the train, validation, and test partitions, with 100K, 18K, and 13K samples.

## 4.2 Consistency of similarity Tasks

**Evaluation Metrics and Setup.** To comprehensively evaluate the performance of our approach in maintaining the consistency of similarity, We design two metrics to evaluate the performance of our model:

1) Consistency of Similarity (SC): To measure whether our method actually captures the similarity relationship between molecules, we directly compare the similarity relationship between the same set of molecules in chemical space and in latent space, and if these two similarities are consistent, then it can be shown that the latent space learns the potential similarity metric of chemical space. In our work, we use Tanimoto Similarity to measure similarity in chemical space and Cosine Similarity to measure similarity in latent space. Considering that Cosine Similarity takes values in the range [-1,1], we normalize Cosine Similarity to [0,1] considering that the two similarity measures should be aligned. Both similarity measures are as large as possible, i.e. 0 means least similar and 1 means most similar. To represent the consistency of these two similarity measures, we introduce Consistency of Similarity, Consistency of Similarity = Tanimoto Similarity / Cosine Similarity. The closer the Consistency of Similarity is to 1 then it can indicate that the Consistency of Similarity is better, which means that our model better maintains the similarity information. To calculate Consistency of Similarity, we randomly sampled 500 pairs of molecules and calculated their tanimoto similarity. The 500 pairs of molecules are then used as input to get the latent variable of these pairs of molecules and to compute their Cosine similarity.

2) Reconstruction and Validity: For a molecular generative model with latent space, reconstruction accuracy and validity are key evaluation metrics. Reconstruction accuracy measures how well the model can embed a molecule into the latent space and then decode it back to its original form, indicating the quality of the learned latent representation. To calculate reconstruction rate, we sample 500 molecules as input. We report the portion of restoring molecules that are identical to the input molecules. Validity, on the other hand, assesses whether the generated molecules are chemically valid, adhering to basic chemical rules. High reconstruction accuracy ensures the model captures the essential features of molecules, while high validity ensures the generated molecules are suitable for practical use in areas like drug discovery. To compute validity, we sample 500 latent vectors from the prior distribution and restore these latent variables to molecules.

3) Uniqueness: A robust molecular generative model should generate unique compounds, which means its latent space includes adequate molecules and it can help to discover new compounds. To calculate uniqueness, we randomly sample 500 latent variables and report the percentages of the unique molecules among all generated molecules.

Table 1: Results of consistency of similarity, reconstruction, validity and uniqueness. For consistency of similarity, a number closer to 1 indicates better consistency. For the other metrics, a higher number indicates a better generation quality.

| Datasets | Methods | Consistency of similarity | Reconstruction (%) | Valid (%) | Uniqueness (%) |
|---|---|---|---|---|---|
| ZINC | TransVAE | 1.61 | 93.6 | 56.6 | 99.4 |
|  | TransVAE+MSCR | **1.04** | **97.3** | **67.8** | 99.4 |
| CHEMBL | TransVAE | 1.73 | 92.1 | 85.0 | 99.6 |
|  | TransVAE+MSCR | **1.08** | **97.7** | **86.4** | **99.8** |
| QM9 | GEOLDM | 2.03 | 72.6 | 93.9 | 98.8 |
|  | GEOLDM+MSCR | **1.34** | **72.8** | **94.0** | **98.9** |

**Results and Analysis.** Table 1 shows that the model adding MSCR outperforms previous original model. For the original latent space, similarity in chemical space and that in latent space is not aligned. In contrast, with the adding of MSCR, the consistency of similarity get an obvious improvement, which means MSCR benefits to help capture similarity relationship in chemical space and maintain it in latent space. Except consistency of similarity, our method achieves varying degrees of improvement on reconstruction, validity and uniqueness. We argue that on the one hand the augmented data from MMPs improves generalization and robustness for the model. The information of similarity provides a way for the model to learn inter-molecular relationships. It allows the model to capture finer-grained information and improves the characterization ability. Thus, while not targeting these three goals, the more general metrics of these molecular generative models achieved gains with the combined improvement in characterization capabilities.

## 4.3 Molecular Optimization Tasks

**Evaluation Metrics and Setup**. To demonstrate the improvement in downstream tasks of molecular design through maintaining consistency of similarity, we utilize our model to produce novel molecules with desired properties based on given molecules (i.e. lead optimization). We consider the following two properties: GSK3$\beta$ and JNK3, which measure the inhibition the scores against glycogen synthase kinase-3$\beta$ and c-Jun N-terminal kinase-3 target proteins,respectively. We apply the following three paradigms of lead optimization :

- Bayesian Optimization (BO): Following Jin et al. (2018), we first train a generative model that associate molecule with a latent variables. Then train a Gaussian Process to predict properties values given its latent variables. Then we perform 5 iterations of BO with upper confidence bound (UCB) as acquisition function.

- Gradient descent (GD): Following Hoffman et al. (2022), we train a generative model with latent space. Then we use approximate gradient descent for optimization. to Every iterations, we randomly sample 50 neighbor latent variables. And then these latent variables are decoded to corresponding molecules, which can be used to calculate approximate gradients. We perform 5 iterations of such approximate gradients descent.

- Random Sampling (RS): Given an initial molecule and trained generative model with latent space, we map the given molecule to the latent space and randomly sample 100 other latent variables around these, decoding the sampled latent variables to obtain desired molecules.

**Results and Analysis.** Following Jin et al. (2020), our evaluation effort measures various aspects of lead optimization. We generate 500 molecules and compute the Success,novelty and diversity. Our results are summarized in Table 2,3,4. As a whole, both for GSK3$\beta$ and JNK3, our model achieves some improvement in the performance of the original method. It is worth noting that the introduction of MSCR for the JNK task do not enhance Novelty when the entire model is trained on the CHEMBL dataset. We believe that the main reason for this is the lack of good initial molecules for JNK3 (740 positives). The number of known molecules with better JNK3 values is very small compared to GSK3$\beta$ (2665 positives). Therefore, in the face of lead optimization, it is difficult for the model to jump out of the original given molecules, so it does not achieve better performance in novelty. In addition, since the methods of BO, GD and RS are still relatively simple, it is difficult to optimize very good molecules in the task of molecular optimization, which is itself very difficult. However, this does not affect our comparison, and it is still very intuitive to see the improvement of the comprehensive ability of the model by adding MSCR. In this section, we did not choose GEOLDM for our experiments, mainly because there is no simple paradigm of lead optimization for 3D molecular optimization, and it is very difficult to optimize the properties of 3D molecules directly. Therefore, if we need to optimize GSK3$\beta$ and JNK3 for GEOLDM, we need to introduce additional guidance, which is beyond the content of this paper.

Table 2: The results on lead optimization with BO on GSK3$\beta$ and JNK3

| Dataset | Methods | BO | | | | | |
| --- | --- | --- | --- | --- | --- | --- | --- |
| | | GSK3$\beta$ | | | JNK3 | | |
| | | Success (%) | Novelty (%) | Diversity (%) | Success (%) | Novelty (%) | Diversity (%) |
| ZINC | TransVAE | 43.4 | 11.6 | 90.1 | 33.5 | 3.9 | 88.2 |
| | TransVAE+MSCR | **47.8** | **24.5** | **93.7** | **38.4** | **4.3** | **88.9** |
| CHEMBL | TransVAE | 46.7 | 11.7 | 91.3 | 35.9 | 7.8 | 88.4 |
| | TransVAE+MSCR | **49.6** | **23.3** | **94.0** | **39.4** | **10.1** | **89.1** |

Table 3: The results on lead optimization with GD on GSK3$\beta$ and JNK3

| Dataset | Methods | GD | | | | | |
| --- | --- | --- | --- | --- | --- | --- | --- |
| | | GSK3$\beta$ | | | JNK3 | | |
| | | Success (%) | Novelty (%) | Diversity (%) | Success (%) | Novelty (%) | Diversity (%) |
| ZINC | TransVAE | 57.4 | 30.3 | 90.8 | 35.3 | 17.8 | 88.7 |
| | TransVAE+MSCR | **60.3** | **40.9** | **93.3** | **40.8** | **23.4** | **90.9** |
| CHEMBL | TransVAE | 57.1 | 31.1 | 91.9 | 35.6 | **18.9** | 89.1 |
| | TransVAE+MSCR | **59.2** | **40.7** | **94.6** | **40.3** | 18.8 | **91.3** |

Table 4: The results on lead optimization with RS on GSK3$\beta$ and JNK3

| Dataset | Methods | RS | | | | | |
|---------|---------|-----|-----|-----|-----|-----|-----|
| | | GSK3$\beta$ | | | JNK3 | | |
| | | Success (%) | Novelty (%) | Diversity (%) | Success (%) | Novelty (%) | Diversity (%) |
| ZINC | TransVAE | 32.3 | 11.6 | 90.2 | 23.3 | 1.7 | 88.1 |
| | TransVAE+MSCR | **40.3** | **20.8** | **90.3** | **32.3** | **2.0** | **90.5** |
| CHEMBL | TransVAE | 33.4 | 12.7 | 90.7 | 24.1 | 1.9 | 89.6 |
| | TransVAE+MSCR | **40.1** | **21.2** | **91.0** | **33.6** | **2.1** | **91.5** |

## 4.4 ABLATION STUDY

In our methods, MSCR consists of two items (i.e distribution terms and metric items). These two items are helpful in capturing similarity and improving modeling capabilities. As shown in Table 5. We can observe that the distribution and metric items are both improving for performance. In general, models with only distribution items perform better than models with only metric items. This suggests that tuning from distribution is a greater enhancement to the model, which is also in line with the original intent of our design, which is to treat distribution items as a coarse-grained tuning and metric items as a fine-grained tuning.

Table 5: The ablation study results of MSCR(only distribution item), MSCR(only metric item) and MSCR

| Datasets | Methods | Consistency of Similarity | Reconstruction (%) | Valid (%) | Uniqueness (%) |
|----------|---------|---------------------------|--------------------|-----------|----------------|
| ZINC | TransVAE+MSCR(only distribution item) | 1.17 | 95.9 | 63.2 | 99.4 |
| | TransVAE+MSCR(only metric item) | 1.23 | 94.3 | 59.7 | 99.4 |
| | TransVAE+MSCR | **1.04** | **97.3** | **67.8** | **99.4** |
| CHEMBL | TransVAE+MSCR(only distribution item) | 1.20 | 96.1 | 85.5 | 99.6 |
| | TransVAE+MSCR(only metric item) | 1.35 | 95.8 | 85.2 | 99.7 |
| | TransVAE+MSCR | **1.08** | **97.7** | **86.4** | **99.8** |
| QM9 | GEOLDM+MSCR(only distribution item) | 1.41 | 72.7 | 93.9 | 98.8 |
| | GEOLDM+MSCR(only metric item) | 1.47 | 72.6 | 93.9 | 98.8 |
| | GEOLDM+MSCR | **1.34** | **72.8** | **94.0** | **98.9** |

## 5 FUTURE WORK AND CONCLUSION

In this paper, we present a plug-in-play regularization that maintains the consistency of chemical space and latent space, MSCR. MSCR enables the model to capture the similarity relationship well from both distribution and metric perspectives, improving the model's capability. In addition we have conducted experiments on multiple datasets and multiple metrics to prove that our method is indeed simple and effective. It is worth noting that our approach aligns the two spaces directly through distributions and metrics, but for the particular problem, these two perspectives can not adequately portray the relationship between the two spaces, and in the follow-up our work is to dig into the underlying relationship between the different spaces to portray the similarity, and propose a more generalized approach.

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
