# OpenReview forum: "Similarity-Driven Regularization for Aligning Chemical and Latent Spaces in Molecular Design"
_ICLR.cc/2025/Conference — ICLR 2025 Conference Withdrawn Submission_

### Official Review · Reviewer_xhDx · 2024-10-18

**Soundness:** 2
**Presentation:** 2
**Contribution:** 2
**Rating:** 3
**Confidence:** 4

**Summary:**

This paper proposes a Molecular Similarity-Aware Consistency Regularization (MSCR) method to address the inconsistency between molecular similarity in the chemical space and that in the latent space in molecular design. The method uses Matched Molecules Pairs (MMPs) for data augmentation and introduces a new loss component to maintain molecular similarity. Experiments on multiple datasets and models show that MSCR can improve the quality of the latent space and the performance of molecular design.

**Strengths:**

1. The issue of inconsistency between chemical and latent spaces in molecular design is important and well-motivated.

2. The visualizations provided in the paper help to highlight molecular inconsistencies and the contributions of the work. This is an important aspect as it allows for a better understanding of the model's behavior and the problem at hand.

3. The proposed MSCR method is a straightforward regularization approach.

4. I appreciate that the paper considers the applicability of the proposed regularization to different generative models from a plug-and-play perspective.

**Weaknesses:**

1. Diffusion-based molecular generation models have become more and more popular than VAE-based and GAN-based approaches since the year 2022, but this paper discusses far less related work in this category, including Pocket2Mol [1], DiffBP [2],  DecompDiff [3], etc.

2. While I appreciate the plug-and-play property of the proposed regularization, only three generative models have been evaluated experimentally, lacking an evaluation on the applicability to the most current molecular design models, especially those based on diffusion.

3. The paper does not discuss well how to define “similarity” in chemical space, which is the most challenging issue. For example, two “structurally” similar molecules may have completely different properties, and two molecules that look completely different may perform the same function. How would the proposed regularization term handle these cases where structural similarity does not correlate with functional similarity? Perhaps a discussion or experiment on this specific aspect will be helpful.

4. While the paper compares MSCR with the base models (TransVAE and GEOLDM), it would be beneficial to include a more comprehensive comparison with other state-of-the-art methods in molecular design

5. There is a lack of discussion on the scalability of the proposed method to larger datasets. Additionally, the computational complexity of the MSCR algorithm could be analyzed to assess its practical applicability.

6. Using consistency loss to constrain the consistency of the input and embedding space is not a new problem; it is a classic problem in the field of manifold learning. However, I have not seen any discussion with these work in this paper. More importantly, augmenting the inputs and then imposing a consistency loss is not fundamentally different from contrastive learning, especially the common graph contrastive learning methods. Considering that there are already so many contrastive learning methods for molecular generation, the technical contribution of this paper is limited.

[1] Peng X, Luo S, Guan J, et al. Pocket2mol: Efficient molecular sampling based on 3d protein pockets[C]//International Conference on Machine Learning. PMLR, 2022: 17644-17655.

[2] Lin H, Huang Y, Zhang O, et al. Diffbp: Generative diffusion of 3d molecules for target protein binding[J]. arXiv preprint arXiv:2211.11214, 2022.

[3] Guan J, Zhou X, Yang Y, et al. DecompDiff: diffusion models with decomposed priors for structure-based drug design[J]. arXiv preprint arXiv:2403.07902, 2024.

**Questions:**

See weaknesses. I will reconsider my score in light of the author's response.

---

### Official Review · Reviewer_nsJZ · 2024-10-18

**Soundness:** 3
**Presentation:** 2
**Contribution:** 3
**Rating:** 5
**Confidence:** 3

**Summary:**

This paper introduces a novel regularization method that aims to align the chemical similarity of novel molecules produced by a generative model to proximity of the representation of these molecules in the model’s latent space. This method attempts to address inconsistencies between similarity in the chemical space and latent space. This is useful to enable confident navigation through a model’s latent space to find molecules that are similar in structure and property to a particular input molecule of interest.

**Strengths:**

The paper clearly outlines the motivation for this exploration, and introduces the new loss function in a piece-wise way that helps the reader understand the intention behind each additional term towards this main goal.

The paper does a good job of evaluation across the three metrics of consistency, reconstruction and validity, and uniqueness. I particularly appreciate the introduction of the consistency metric.

Nice ablation statistics in Tables 1 and 2, and I appreciate the MSCR ablation studies in Table 5.

**Weaknesses:**

It would be helpful to describe more robustly at the beginning of the paper what is meant by similarity. “Chemical similarity” is a vague term that can range from structural similarity to similar chemical properties. There is a version of this paper that optimizes for ADMET, for example, which would be a very different kind of evaluation.

The paper chooses two baselines, TransVAE and GEOLDM, and thereby tests a transformer-based and autoencoder+diffusion-based latent space It would be helpful to choose another model with a substantively different architecture to show that this method generalizes to substantially different model architectures. Also, there are other models, like MolMIM, that already attempt to solve the chemical similarity problem, so it would be imperative to compare against such a model to demonstrate the additional value added by MSCR.

**Questions:**

- Maybe I missed this, but what model is being used for Figure 1?
- How is chemical similarity defined (e.g. how do you calculate a score of, say, 0.8, in Figure 1b? Is this a metric with a precedent, and if yes, can you please cite? If you are defining your own metric, please specify its calculation.
- Why did you choose to define distribution consistency in a way that’s not symmetric (using the KL) versus using something like Jensen Shannon?
- While there is precedent to a two-stage training approach, because of what you’re trying to do here, I am surprised that it outperforms one-stage. I’d be curious to see more results or information as to why this is the better training regime.
- Interpolation results — we begin the paper using Figure 1 to highlight the problem, but we don’t redo this analysis for any of the models + MSCR. I want to see a depiction of the latent space and, ideally, an interpolation between molecules in the latent space, to demonstrate that the space itself is more robustly chemically consistent.

---

### Official Review · Reviewer_UAbj · 2024-11-02

**Soundness:** 2
**Presentation:** 2
**Contribution:** 2
**Rating:** 3
**Confidence:** 4

**Summary:**

This paper presents a plug-in-play regularization that maintains the consistency of chemical
space and latent space, MSCR. MSCR enables the model to capture the similarity relationship from both distribution and metric perspectives, improving the model’s capability. In addition, the authors have conducted experiments on multiple datasets and multiple metrics.

**Strengths:**

The experiments are complete.

**Weaknesses:**

1. The introduction is too lengthy. I kindly suggest the authors focus on the latent space regulization problem and just give a brief introduction to the latent space generative model
2. The introduction to the Molecular Similarity Consistency issue is overly verbose and convoluted. It would be clearer to directly state that maintaining the equivalence of latent similarity and molecular similarity is crucial for optimization in molecule optimization and editing.
3. The Matcherd molecules paris are not novel. The authors also don’t indroduce novel methods for creating MMPs.
4. The MOLECULAR SIMILARITY-AWARE CONSISTENCY REGULARIZATION is not novel. Minimizing the KL divergence between the encoder’s  output and the output of its semantically preserved transformation or forcing the similarity between the sample pairs and latent pairs are common technique as regulization.
5. The improvements on generation and reconstruction performance are very maginal, especially on Diffusion Model.

**Questions:**

1. How to excute MATCHED MOLECULES PAIRS? Please introduce more details
2. Could you provide some case studies and visualization of the generated and optimized molecules?

---

### Official Review · Reviewer_uDQp · 2024-11-03

**Soundness:** 2
**Presentation:** 1
**Contribution:** 1
**Rating:** 1
**Confidence:** 3

**Summary:**

This paper presented a plug-and-play similarity-driven regularization method, named Molecular Similarity-Aware Consistency Regularization (MSCR), to align chemical space and latent space. The authors assume that similar molecules should map to similar latent variables.
Matched Molecule Pairs (MMPs) are introduced to serve as a more robust augmentation method.
Based on such augmentation, straightforward regularization terms are applied to training variational autoencoders from both the perspectives of metrics and distribution.
Experiments show that the proposed methods can make the latent embeddings of similar molecules maintain similar and also improve the performance on downstream tasks, such as molecular optimization based on latent space.

**Strengths:**

The inclusion of Figures 1 and 2 significantly enhances the clarity of the paper. These illustrations effectively depict both the underlying motivation and the proposed methodology, providing the reader with a clear and concise understanding of the concepts and approaches discussed. This visual representation aids in making the paper's arguments more accessible and comprehensible.

**Weaknesses:**

The reproducibility is not ensured.

The motivation presented in the paper appears to be somewhat lacking from a chemistry perspective. Assessing molecular similarity is an open challenge, as minor structural changes in an atom can significantly alter a molecule's functions and properties. The authors could benefit from referencing well-known concepts such as activity cliffs (ACs) to strengthen their argument.

The use of Tanimoto Similarity as the similarity metric in chemical space, while straightforward, may not adequately capture the true 'chemical' similarity.

The experimental evaluation needs to be more comprehensive. For instance, proposed regularization terms are introduced alongside the original VAE training losses, suggesting the presence of a hyperparameter to control regularization strength. However, there is an absence of ablation studies addressing this aspect.

Furthermore, while multiple methods exist for measuring molecular similarity, only one approach is utilized here, lacking post-hoc analysis. A deeper analysis is necessary to understand how increased consistency enhances the performance of Bayesian optimization.

In some cases, the reported improvements, such as with GeoLDM on QM9, are too marginal.

Overall, the presentation falls short of academic standards, with several typographical errors present.

Some related references are not cited in the paper. There are lots of related works that focus on semantics in latent space, though most of them are in the field of computer vision.

**Questions:**

See the weaknesses above.

**Details Of Ethics Concerns:**

No ethics concerns.

---

### Note · Authors · 2024-11-19

I have read and agree with the venue's withdrawal policy on behalf of myself and my co-authors.